# Incubation Time Influences Organic Anion Transporter 1 Kinetics and Renal Clearance Predictions

Aaron O. Buaben [1] and Ryan M. Pelis [2,*]

[1] Department of Pharmacology, Dalhousie University, Halifax, NS B3H 4R2, Canada
[2] Drug Disposition, Pharmacokinetic Sciences, Novartis Institutes for Biomedical Research, Cambridge, MA 02139, USA
* Correspondence: ryan.pelis@novartis.com

**Abstract:** Accurate predictions of drug uptake transporter involvement in renal excretion of xenobiotics require determination of in vitro transport kinetic parameters under initial-rate conditions. The purpose of the present study was to determine how changing the incubation time from initial rate to steady state influences ligand interactions with the renal organic anion transporter 1 (OAT1), and the impact of the different experimental conditions on pharmacokinetic predictions. Transport studies were performed with Chinese hamster ovary cells expressing OAT1 (CHO-OAT1) and the Simcyp Simulator was used for physiological-based pharmacokinetic predictions. Maximal transport rate and intrinsic uptake clearance ($CL_{int}$) for PAH decreased with increasing incubation time. The $CL_{int}$ values ranged 11-fold with incubation times spanning from 15 s ($CL_{int,15s}$, initial rate) to 45 min ($CL_{int,45min}$, steady state). The Michaelis constant ($K_m$) was also influenced by the incubation time with an apparent increase in the $K_m$ value at longer incubation times. Inhibition potency of five drugs against PAH transport was tested using incubation times of either 15 s or 10 min. There was no effect of time on inhibition potency for omeprazole or furosemide, whereas indomethacin was less potent, and probenecid (~2-fold) and telmisartan (~7-fold) more potent with the longer incubation time. Notably, the inhibitory effect of telmisartan was reversible, albeit slowly. A pharmacokinetic model was developed for PAH using the $CL_{int,15s}$ value. The simulated plasma concentration-time profile, renal clearance, and cumulative urinary excretion-time profile of PAH agreed well with reported clinical data, and the PK parameters were sensitive to the time-associated $CL_{int}$ value used in the model.

**Keywords:** kidney; drug transport; organic anion transporter 1; physiological-based pharmacokinetics; Mechanistic Kidney model; renal clearance; drug–drug interaction

## 1. Introduction

The kidney is important for eliminating from the body a variety of small molecular weight therapeutic drugs "xenobiotics" carrying either a net-positive (cationic drugs) or negative (anionic drugs) charge at physiological pH [1]. For many of these drugs, their urinary elimination is accomplished by glomerular filtration as well as active tubular secretion. Indeed, of the top 200 prescribed drugs in the United States in 2010, 32% are eliminated predominantly by the kidney, and of these, 92% are actively secreted by renal tubules [2]. Renal tubular drug secretion involves the concerted activity of uptake transporters at the basolateral membrane and efflux transporters at the apical membrane of renal tubule cells [1]. Tubular secretion is an important determinant of plasma drug concentration, is a site of drug–drug interactions (DDIs), and for select drugs, their accumulation into proximal tubule cells mediated by transporters can lead to nephrotoxicity [2,3]. Accordingly, the US Food and Drug Administration (US-FDA) and European Medicines Agency (EMA) in their guidance on drug interaction studies for the pharmaceutical industry recommend determining in vitro the interaction of new molecular entities (NMEs) with several renal drug transporters [4,5].

For anionic drugs that undergo active tubular secretion, their uptake into renal tubule cells is mediated predominately by the organic anion transporter 1 (OAT1) and the organic anion transporter 3 (OAT3) [6]. A number of transporters at the apical membrane are considered candidates for mediating the efflux of anionic drugs into the tubule lumen, including the multidrug resistance-associated proteins 2 and 4 [7]. Of the top 200 prescribed drugs in the United States in 2010, 28% interact with OAT1 and 41% with OAT3, highlighting the importance of these organic anion transporters in renal drug elimination and DDIs [2].

Regulatory agencies have outlined recommendations on when and how to test whether an NME is a substrate and/or an inhibitor of renal drug transporters. First, for inhibition studies, and for determining if NMEs are substrates, cultured cells with heterologous expression of an individual transporter are advocated [5,8,9]. If the in vivo renal clearance of an NME is a substantial component of its total clearance, and there is evidence of its active tubular secretion, it is recommended to test if it is a substrate of OAT1 and OAT3—assuming that the NME has the appropriate physicochemical properties to be a substrate, i.e., a hydrophilic small-molecular-weight organic anion [2]. Given the broad ligand selectivity of OAT1 and OAT3, and their potential involvement in DDIs, it is recommended for all NMEs to determine if they inhibit OAT1/3 transport activity [5].

Once an NME is determined a substrate and/or inhibitor, the extent of its interaction with the transport protein is determined using in vitro kinetic transport studies. For substrates, this entails determining the kinetic parameters of uptake, i.e., maximal transport rate ($J_{max}$) and the Michaelis constant ($K_m$), and for inhibitors, the inhibition potency of probe substrate uptake (inhibition constants, $IC_{50}$ or $K_i$ value) [8]. The kinetic parameters can then be used in either static or dynamic mechanistic models (i.e., physiological-based pharmacokinetic (PBPK) models) to predict the involvement of transporters in pharmacokinetics (PK) of NME substrates, and potential DDIs that may occur at the transporters due to inhibition caused by perpetrator drugs. Ultimately, the output of these models is used to guide the clinical development of NMEs [5].

Prior to performing kinetic transport experiments, the time course of uptake should be determined to establish the linearity of uptake, and it is recommended to use kinetic parameters determined at an initial-rate time point in downstream pharmacokinetic predictions [8,10]. To our knowledge, there have been no dedicated studies in the literature to show what impact a changing incubation time has on the uptake transporter kinetics and subsequent PK predictions. Thus, the present study examined the kinetics of PAH (a prototypical OAT1 substrate) transport by OAT1 over a wide range of incubation time-points, from initial rate to approximate steady state. The kinetic parameters obtained at the different incubation time points were then input into the Mechanistic Kidney model (MechKiM) within the Simcyp Simulator to determine how the time point used influences PK predictions.

## 2. Materials and Methods

### 2.1. Reagents and Chemicals

[$^3$H]-*Para*-aminohippurate (40 Ci/mmol) was obtained from American Radiochemicals (St. Louis, MO, USA). F12 Kaighn's modification medium, fetal bovine serum (certified, U.S. origin), 1% penicillin-streptomycin solution, and hygromycin B were purchased from ThermoFisher Scientific (Waltham, MA, USA). All other chemicals were of the highest purity possible and were purchased from Sigma Aldrich (St. Louis, MO, USA).

### 2.2. Cell Culture

The cloning of the human ortholog of OAT1 from the human kidney and its stable expression in Chinese Hamster Ovary Flp-In cells (CHO-OAT1) was described previously [11]. CHO-OAT1 cells were cultured in a complete medium (F12 Kaighn's modification medium, 10% fetal bovine serum and 1% penicillin-streptomycin) supplemented with hygromycin B

(200 µg/mL final concentration) in a humidified atmosphere of 5% $CO_2$:95% air at 37 °C. All cells were grown to confluence in 24-well flat bottom plates for transport experiments.

*2.3. Transport Experiments*

Transport experiments were conducted using room temperature Waymouth's buffer (WB) containing [$^3$H]-labeled *para*-aminohippurate ([$^3$H]*PAH*; 20 nM), and in some cases, either unlabeled PAH or inhibitor drug. The chemical composition of the WB was (in mM): 135 NaCl, 5 KCl, 28 D-glucose, 1.2 MgCl$_2$, 2.5 CaCl$_2$, 0.8 MgSO$_4$$^{2-}$, and 13 hydroxyethyl piperazineethanesulfonic acid (HEPES) (pH 7.4). Transport experiments involved aspirating the media, rinsing the cells quickly once with WB (0.4 mL), and then adding the transport solution (0.4 mL) to the wells for an amount of time as indicated in the figure legends. Uptake was stopped by aspirating the transport buffer and rinsing the wells three times with ice-cold WB (0.4 mL). The cells were lysed in 0.5 N NaOH/1% sodium dodecyl sulfate (0.4 mL) for 30 min on an orbital shaker and the lysates neutralized with 1 N HCl (0.2 mL). Radioactivity content was measured by liquid scintillation counting (Tri-Carb® 2910TR LSA; PerkinElmer, Waltham, MA, USA). Several wells not used for transport were used for determining the number of cells per well with a hemocytometer. In our hands, the uptake of [$^3$H]*PAH* into CHO-OAT1 cells is typically around 10-fold greater than the uptake into CHO parental cells when using an initial rate time point (15 s).

To establish initial rate and time-to-steady state, the time course of PAH uptake into CHO-OAT1 cells was determined using four different concentrations of PAH (0.02, 10, 50 or 100 µM) with uptake at each concentration measured at 15 s, 30 s, 1 min, 2 min, 5 min, and 10 min. The time-dependent uptake of PAH at each concentration tested was fit to a curve that describes the pseudo-first-order association kinetics of the interaction between ligand and receptor (i.e., PAH and OAT1 in this case):

$$Y = Y0 + (Plateau - Y0) * (1 - \exp(-K * X))$$

where $Y$ is the PAH uptake, $Y_0$ is the PAH uptake value at time = 0, K is the rate constant, and X is time.

The saturable kinetics of PAH uptake into CHO-OAT1 cells was determined using an equation that describes the competitive inhibition of radiolabeled substrate uptake by increasing concentrations of unlabeled substrate:

$$J = \frac{J_{max}[*PAH]}{K_m + [PAH]} + C$$

*J* is the rate of [$^3$H]*PAH* uptake from a concentration of [$^3$H]*PAH* in the transport solution equal to [*PAH*] (20 nM). [*PAH*] is the concentration of unlabeled *PAH* in the transport solution. *C* is defined as the non-saturable component of [$^3$H]*PAH* uptake that is most likely due to factors such as non-specific binding, incomplete rinsing, and passive diffusion. *J$_{max}$* is the maximal transport rate, and *K$_m$* is the Michaelis constant, i.e., the *PAH* concentration resulting in half-maximal transport.

To examine inhibition potency (IC$_{50}$ value determination), the transport solution contained a fixed concentration of [$^3$H]*PAH* (20 nM) and increasing concentrations of OAT1 inhibitor (probenecid, furosemide, indomethacin, omeprazole or telmisartan). As above, transport inhibition experiments involved aspirating the media, rinsing the cells quickly once with WB (0.4 mL), and then adding the transport solution (0.4 mL) to the wells for an amount of time as indicated in the figure legends. The concentration of drug to inhibit uptake by 50% (IC$_{50}$ value) was determined by nonlinear regression analysis using the following relationship:

$$J = \frac{J_{app}[*PAH]}{IC_{50} + [I]} + C$$

*J* is the rate of [³H]*PAH* uptake from a concentration of [³H]*PAH* in the transport solution equal to [*PAH*]. [*I*] is the concentration of inhibitor in the transport solution. *IC$_{50}$* is the inhibitor concentration required to reduce the substrate uptake by 50%. *J$_{app}$* is the apparent maximal transport rate, and *C* has the same meaning as noted above.

Also examined was the effect of increasing incubation time on transport inhibition caused by telmisartan, and the time-to-recovery of PAH transport activity following telmisartan removal. In these experiments, the cells were pre-incubated with telmisartan, but telmisartan was not included in the uptake solution. To examine the time-dependent nature of telmisartan inhibition, CHO-OAT1 cells were pre-incubated with telmisartan at four different telmisartan concentrations (10 nM, 50 nM, 100 nM or 200 nM, diluted in WB) for 1, 5, 10 or 30 min. After the pre-treatment period, the cell layer was rinsed rapidly with WB, followed by immediate measurement of [³H]*PAH* uptake for 15 s.

The recovery of PAH transport following pre-treatment of the cells for 30 min with telmisartan (0.5 μM) was investigated in a separate set of experiments. This was done to examine the reversibility of telmisartan inhibition after prolonged incubation periods. We have previously shown reversibility of PAH transport by OAT1 following a very brief (10 s) exposure to telmisartan [12]. Given the hydrophobicity of telmisartan, and the drug's high propensity for protein binding, following telmisartan exposure, the cells were rinsed and incubated in WB containing 10% FBS for 2, 5, 10, 20 or 30 min prior to measuring the uptake of [³H]*PAH* for 15 s.

### 2.4. PBPK-MechKiM Model Development for PAH

The MechKiM within the Simcyp Simulator version 15 (Certara, Inc., Princeton, NJ, USA) was used for simulating the plasma concentration-time profile, cumulative urinary excretion-time profile, and renal cell concentration-time profile of PAH. A DDI trial was also performed using the probenecid compound file contained within the Simcyp Simulator. The GetData Graph Digitizer 2.26 was used to extract clinical data from the literature [12,13]. Further details of the PAH PBPK-MechKiM model can be found in Supplemental Figures S1 and S2.

All simulations were performed using the default "healthy volunteers" population (ages 20–50). For each simulation, ten trials were performed with each trial containing ten subjects, with a ratio of 50% male and 50% female. As with the published clinical studies [12,13], in the virtual trials, PAH was administered as a single intravenous bolus dose of 10 mg · kg$^{-1}$, and plasma sampled over a several hour period. Like the published trials, the first plasma sample in the virtual trials was not taken until 5 min after PAH administration. For the DDI trial, we first gave a single oral dose of probenecid (1 g), and then, three hours later, a single intravenous bolus dose of PAH (10 mg·kg$^{-1}$)—the timing of the PAH administration was set to match the t$_{max}$ of probenecid.

### 2.5. Statistical Analysis

In vitro data are reported as mean ± standard error of the mean. The number of observations (noted in the legends) are the number of times the experiment was conducted in triplicate on a batch of cells at a different passage number. The effect of incubation time on J$_{max}$, K$_m$ and CL$_{int}$ values, and on the time-dependent inhibitory effect of telmisartan, was determined by One-Way Analysis of Variance. Comparison of IC$_{50}$ values at an incubation time of 15 s vs. 10 min was done using a two-tailed unpaired Student's *t*-test. Statistical significance was set at the *p* < 0.05 level. All nonlinear regression and statistical analysis were done with GraphPad Prism (version 7; GraphPad Software, La Jolla, CA, USA).

## 3. Results

### 3.1. Time Course of PAH Uptake into CHO-OAT1 Cells

To establish the time course of PAH uptake, CHO-OAT1 cells were incubated with four different concentrations of PAH (0.02, 10, 50 or 100 μM) using six different incubation time points (15 s, 30 s, 1 min, 2 min, 5 min, and 10 min) (Figure 1). Not surprisingly, when

comparing individual time points, there was an increase in the amount of PAH accumulated as the PAH concentration in the incubation media increased. For example, the uptake at the 10 min time point was >1000-fold higher at a bath concentration of 100 μM vs. 0.02 μM. The uptake was approximately linear for 2 min at all PAH concentrations tested, and in each case, the uptake did not achieve steady state by 10 min. The extrapolated time-to-plateau (i.e., steady state) was 38, 41, 49, and 68 min at bath concentrations of 0.2, 10, 50, and 100 μM, respectively. For subsequent saturation kinetic experiments, a time point of 2 min or less was assumed to be an initial-rate time point, while a 45 min incubation was assumed approximate steady state.

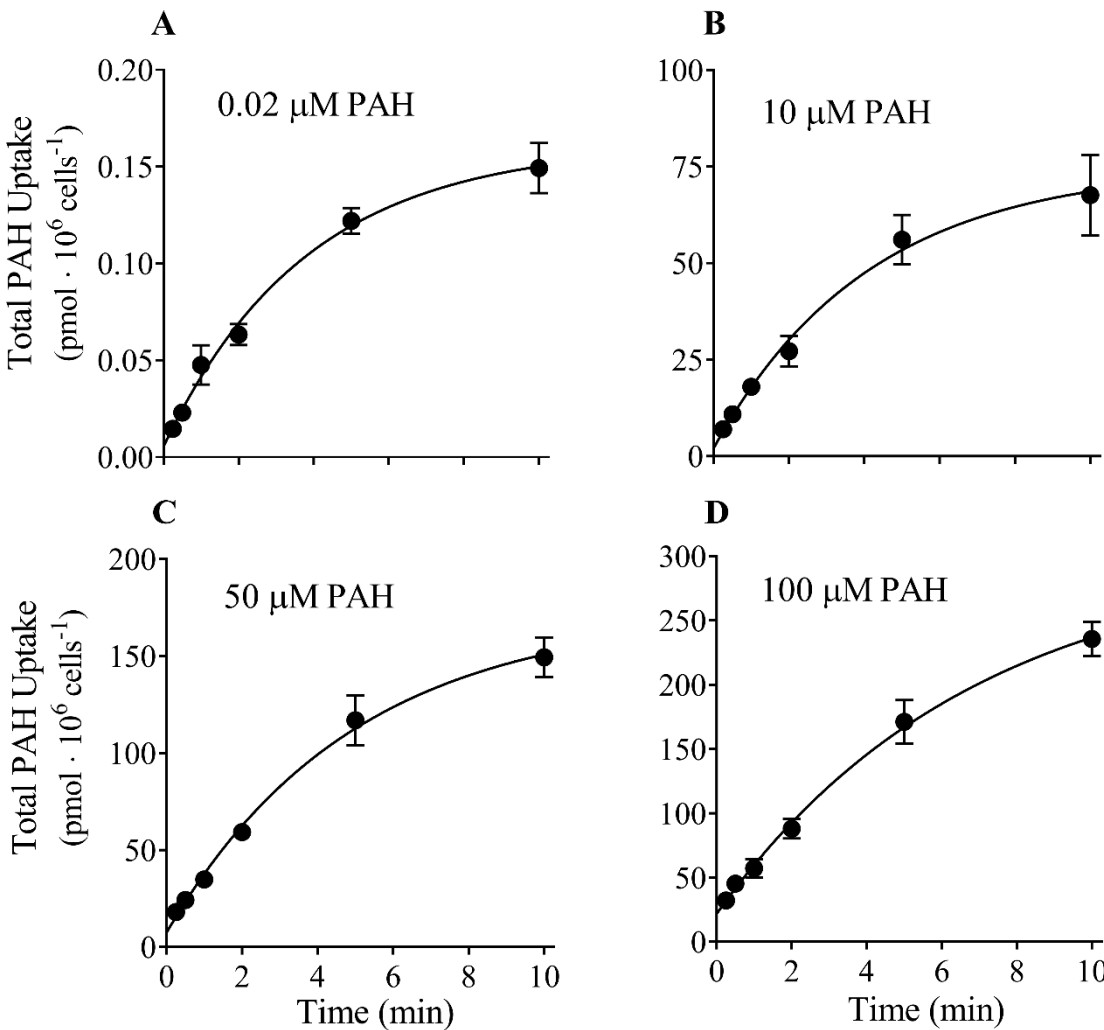

**Figure 1.** Time course of PAH uptake into CHO-OAT1. PAH uptake (total uptake, i.e., radiolabeled + unlabeled) at different incubation times was determined using four different concentrations of PAH: 0.02 μM (**A**), 10 μM (**B**), 50 μM (**C**), and 100 μM (**D**). At all four PAH concentrations, the uptake was nearly linear for 2 min, and approached steady state at 10 min. Data are expressed as mean ± SEM of three experiments.

### 3.2. Effect of Time on Transport Kinetics

Figure 2 shows the effect of increasing incubation time (15 s to 45 min) on the $J_{max}$, $K_m$, and $CL_{int}$ ($J_{max}/K_m$) values. There was a significant decrease in the $J_{max}$ value with time (Figure 2A and Supplementary Table S1A). There was also a significant effect of incubation time on the $K_m$ value, where it appeared to be higher at longer incubation times (Figure 2B and Supplementary Table S1A). Due to changes in both $J_{max}$ (decrease) and $K_m$ (apparent increase) values, $CL_{int}$ decreased significantly with increasing time (Figure 2C

and Supplementary Table S1A). The $CL_{int}$ decreased 11-fold when using incubation times ranging from 15 s to 45 min.

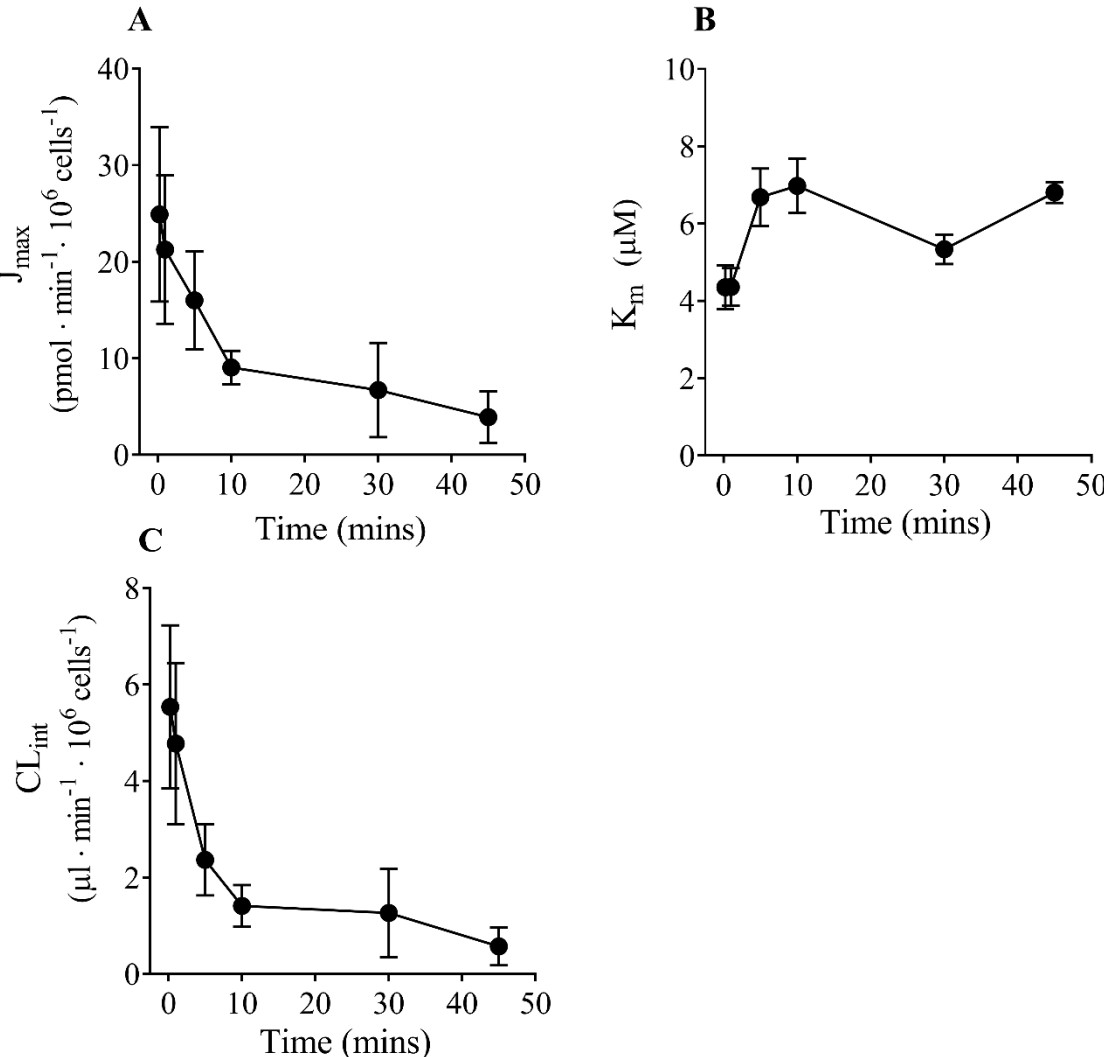

**Figure 2.** Effect of increasing time on $J_{max}$, $K_m$, and $CL_{int}$ values. The kinetic constants obtained at each time point are summarized in Supplementary Table S1A. There was a significant decrease in the $J_{max}$ value with increasing time ($p < 0.05$) (**A**). There was also a significant effect of time on the $K_m$ value ($p < 0.01$), where it appeared to increase with increasing time (**B**). Due to a decrease in the $J_{max}$ value and an apparent increase in the $K_m$ value, there was a significant decrease in $CL_{int}$ as time increased ($p < 0.01$) (**C**). Data are the mean $\pm$ SEM of four experiments. Significant differences between kinetic parameters determined at different time points (15 s to 45 min) were determined by one-way ANOVA.

### 3.3. Effect of Time on Inhibition Potency

Given the significant effect of time on the $K_m$ value, the influence of time on the potency with which five different drugs (probenecid, furosemide, indomethacin, omeprazole, telmisartan) inhibit the 15 s versus 10 min uptake of PAH by OAT1 was further investigated (Figure 3 and Supplementary Table S1B). Inhibition potency for omeprazole and furosemide did not change significantly with time. However, inhibition potency for the remaining three drugs were sensitive to the time of incubation. For indomethacin, inhibition potency was ~2-fold lower at 10 min compared to 15 s, whereas inhibition potency of probenecid was ~2-fold higher at the longer incubation time. Telmisartan was ~7-fold more potent when performing uptake at 10 min versus 15 s.

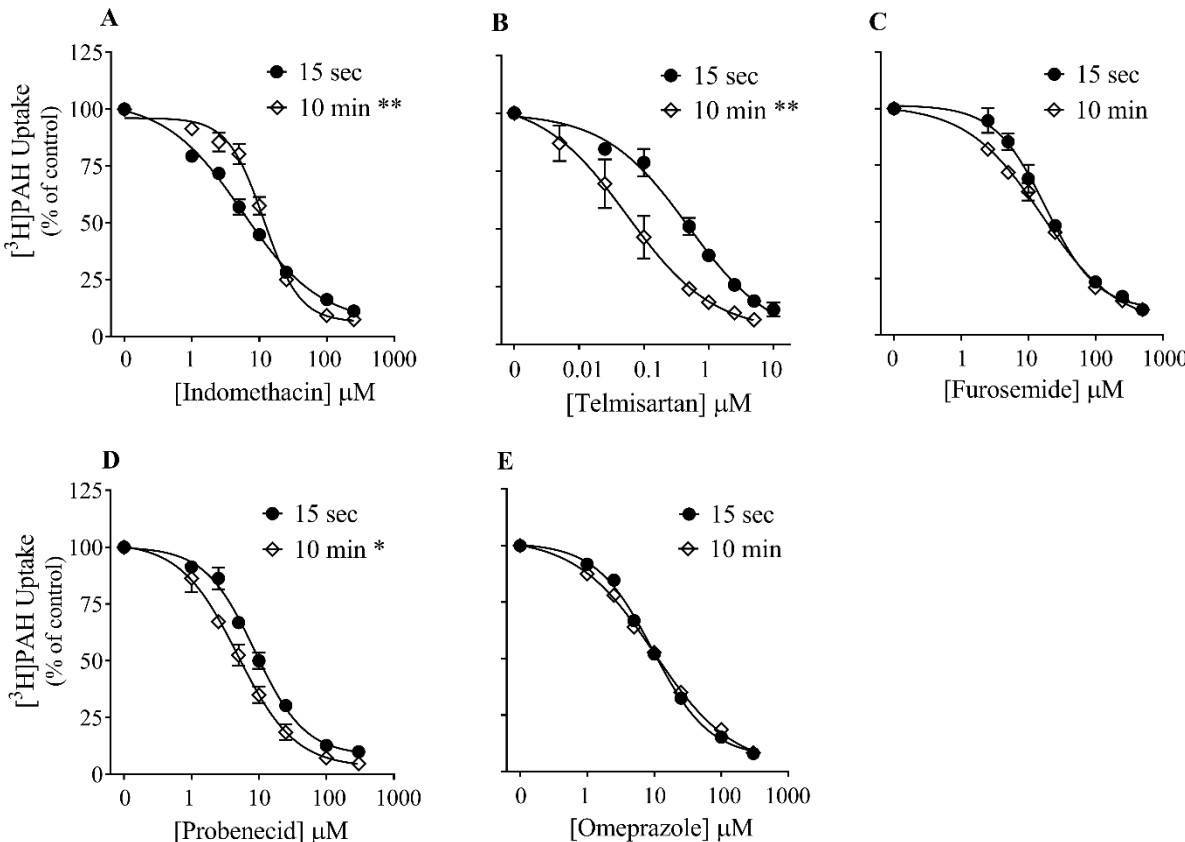

**Figure 3.** Effect of time on inhibition potency. Inhibition of the 15 s versus 10 min uptake of [³H]*PAH* into CHO-OAT1 cells was determined in the presence of increasing concentrations of selected OAT1 inhibitors. $IC_{50}$ values are summarized in Supplementary Table S1B. Data are the mean $\pm$ SEM of four observations and are expressed as a percentage of the control uptake done in the absence of the inhibitor. * $p < 0.05$ and ** $p < 0.01$ indicates a significant difference between $IC_{50}$ values at 15 s vs. 10 min, unpaired Student's *t* test. (**A**) Indomethacin, (**B**) Telmisartan, (**C**) Furosemide, (**D**) Probenecid, and (**E**) Omeprazole.

The relatively large increase in inhibition potency with time caused by telmisartan led us to explore in more detail the time-dependent nature of inhibition. The data from experiments shown in Figure 4 are those that would typically be done to examine whether a drug is a time-dependent inhibitor of drug-metabolizing enzymes. That is, we examined the effect on OAT1 transport activity of increasing concentrations of telmisartan following increasing pre-incubation times. At all four telmisartan concentrations tested, there was a significant effect of time on transport activity, where it decreased with increasing pre-incubation times, from 1 min to 30 min. However, at each concentration tested, there was a significant difference in [³H]*PAH* uptake from 1 min to 5 min, but no difference between the 5, 10 or 30 min time points.

To test the reversibility of telmisartan inhibition after long incubations, the cells were pre-incubated for 30 min with a concentration of telmisartan expected to inhibit nearly all transport activity, followed by its removal and washout for various periods of time before the transport measurement. In a single experiment, there was near-complete recovery of PAH transport after 20 min, indicating that the inhibitory effect of telmisartan is reversible, albeit slowly (Supplemental Figure S3).

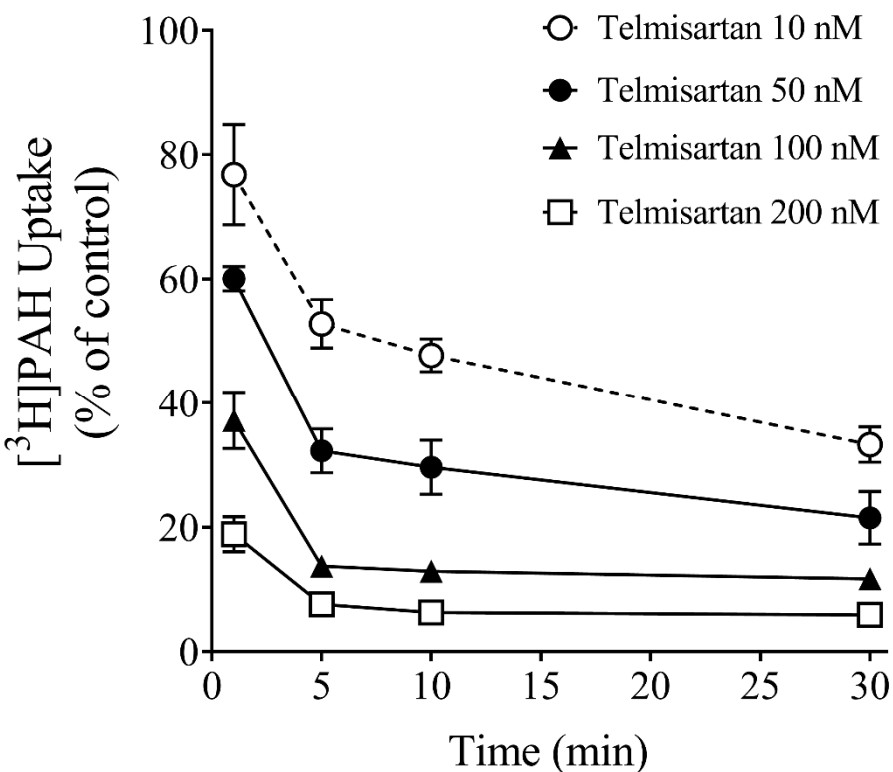

**Figure 4.** Effect of pre-incubation time on OAT1 inhibition by telmisartan. CHO-OAT1 cells were pre-incubated with telmisartan at four different concentrations (10 nM, 50 nM, 100 nM or 200 nM) for 1, 5, 10 or 30 min, followed by the measurement of [³H]*PAH* uptake for 15 s. There was a significant effect of time on PAH uptake at all four concentrations tested: 10 nM ($p < 0.01$), 50 nM ($p < 0.001$), 100 nM ($p < 0.001$), and 200 nM ($p < 0.01$), one-way ANOVA. At each concentration tested, there was a significant difference in [³H]*PAH* uptake from 1 min to 5 min ($p < 0.05$, Tukey's HSD), but no difference between the 5, 10 or 30 min time points ($p > 0.05$, Tukey's HSD).

### 3.4. PBPK Model Predictions

The PBPK-MechKiM model was used to show how differences in transport kinetics due to changing incubation time influences PK predictions. The $CL_{int}$ values obtained at the different incubation time points were used to simulate the plasma concentration-time, urinary excretion-time, and the renal tubule cell concentration-time profile of PAH. The simulated renal clearance was 52 L/h and 18.5 L/h when using $CL_{int}$ values obtained at the 15 s and 45 min incubation time points, respectively. When using the $CL_{int,15s}$, the $AUC_{0-\infty}$ was 3.2-fold lower than when using the $CL_{int,45min}$ (Figure 5A). After 2 h, the percentages of the PAH dose excreted in the urine were 100% ($CL_{int,15s}$) versus 81% ($CL_{int,45min}$) (Figure 5B). The maximum intracellular PAH concentration in the tubule cells was 3.4-fold higher when using $CL_{int,15s}$ versus $CL_{int,45min}$ (Figure 5C).

Next, a DDI trial between PAH and probenecid (administered as a 1 g oral dose) was simulated using the default $K_i$ value within Simcyp for probenecid against OAT1 (4 µM), which is in line with the $IC_{50}$ value that we observed for the PAH uptake inhibition following a 10 min incubation (5.1 µM). In the presence of probenecid, the $AUC_{0-\infty}$ was 2.1-fold higher compared to in its absence. Plasma clearance was reduced 2.0-fold, which resulted in its prolonged half-life (1.9-fold longer). Consistent with these simulated data, co-administration of probenecid (1 g oral dose) caused a 1.8-fold reduction in the plasma clearance of PAH in human subjects [14]. This observation further increased confidence in the OAT1 PBPK-MechKiM model.

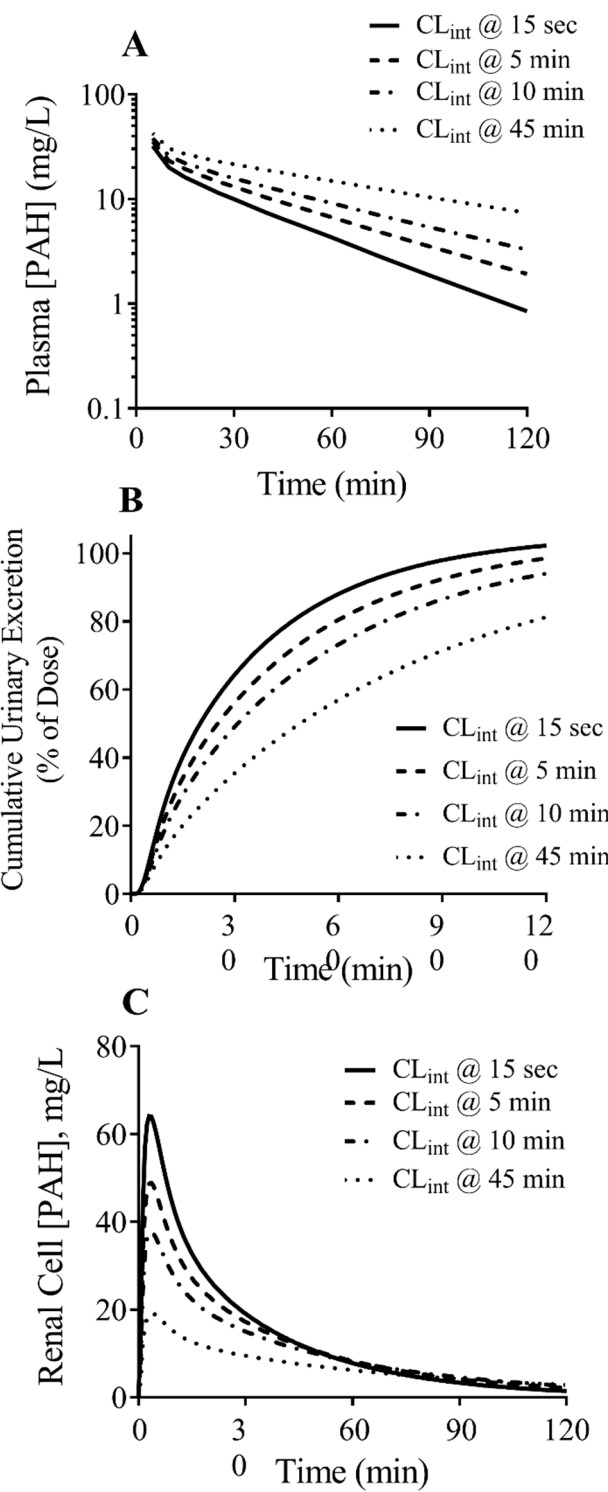

**Figure 5.** Effect of OAT1 $CL_{int}$ differences on simulated pharmacokinetic predictions for PAH using MechKiM. $CL_{int}$ values obtained at the different incubation time points were used to simulate the plasma concentration-time profile (**A**), the cumulative urinary excretion-time profile (**B**), and the renal cell concentration-time profile of PAH (**C**) following an intravenous 10 mg/kg bolus dose of PAH. The simulated plasma concentration-time profile of PAH was optimized to the observed clinical data using the $CL_{int}$ value obtained at 15 s and parameter estimation of the RAF value (see Supplemental Data).

## 4. Discussion

For drug uptake transporters such as OAT1, the kinetic parameters of transport, $J_{max}$ and $K_m$, which define the in vitro intrinsic uptake clearance of a substrate, are essential input parameters required for mechanistic dynamic PBPK modeling, such as with MechKiM [15]. Determination of kinetic parameters for uptake transporters are routinely done in vitro using heterologous expression systems, typically under initial-rate conditions, where a transport measurement is conducted at a single time point where the substrate uptake is known to be in the linear range [8,10]. These conditions are also referred to as "zero-*trans*" conditions, since upon initiation of uptake, the substrate concentration inside of the cell is zero [16]. The purpose of the present study was to examine how a broad range of incubation times from initial rate to approximate steady state influences kinetic parameters of transport for OAT1, and how the time point chosen influences subsequent downstream modeling predictions of its involvement in the renal clearance of the prototypic OAT1 substrate, PAH.

Time course experiments were conducted to establish initial-rate and steady-state time points to use in subsequent saturation kinetic experiments, and these times were estimated to be less than two minutes and 45 min, respectively. The effect of incubation time (15 s–45 min) on the kinetics of PAH transport was examined next. As incubation time increased, there was a decrease in the $J_{max}$ and $CL_{int}$ values. Total substrate uptake is due to the combined influence of passive diffusion and transporter-mediated uptake. The decrease in both kinetic parameters ($J_{max}$ and $CL_{int}$) with increasing time is best explained by differences in the contributions of transporter-mediated uptake and passive diffusion to total PAH flux. As incubation time increased, transport reaches a zero-order flux, where the rate is independent of the substrate concentration, but dependent on time—hence, the decrease in $J_{max}$. At steady state, there is no net flux of substrate across the lipid bilayer, i.e., substrate uptake and efflux are equal. If $CL_{int}$ is estimated beyond the linear phase, it decreases as the transporter-mediated component of uptake becomes diluted by self-exchange and passive diffusion. Under these conditions, the transporter-mediated component of uptake clearance is under-estimated.

The next step was to determine how incubation time-based changes in $CL_{int}$ influence renal distribution and clearance predictions for PAH by PBPK-MechKiM. PBPK models are being increasingly used in a mechanistic manner to define transporter involvement in PK and DDIs [17–19]. However, the predictive performance of mechanistic models for transporters has not been well established, and lag behind those for metabolizing enzymes [5,19]. This is in part due to the lack of understanding of the physiologically-relevant scaling factors for IVIVE, which for OAT1, include differences in its abundance and activity in the in vitro test system compared to in vivo, its distribution along the nephron, the number of renal tubule cells per gram kidney, and the surface area of the basolateral and apical membranes of renal tubule cells [20]. In addition, accurate predictions require that the experimental in vitro conditions chosen for estimating kinetic parameters are reflective of the kinetics of drug transport in vivo.

Here, we established a model to describe the renal clearance of PAH using the in vitro $CL_{int}$ value obtained at initial rate (15 s) and parameter estimation of the REF value, thereby accounting for unknowns in the relevant scaling factors for OAT1 for IVIVE. Using this model, our predicted renal clearance of PAH agreed reasonably well with the observed clinical data. We then performed simulations using $CL_{int}$ values obtained at increasing incubation times out until a time point that would be assumed steady state (45 min), where the uptake $CL_{int}$ of PAH is attributed to self-exchange and passive diffusion. There were ~3-fold differences in the renal clearance predictions of PAH when using $CL_{int,15s}$ vs. $CL_{int,45min}$. Thus, it can be expected that as the incubation time point used for in vitro uptake kinetic experiments exceeds an initial rate time point, the predicted contribution of mediated transport to renal clearance would be underestimated due to the increasing contribution of passive diffusion to the overall intrinsic clearance in vitro. It is important to note that the assumption of using an initial-rate time point for determining in vitro

transport kinetic parameters may be best for predicting PK following single-dosing as done for PAH here. However, under multiple dosing regimens, where steady state PK is attained, other in vitro approaches for estimating transport kinetics may be more appropriate, such as equilibrium exchange kinetics (work currently ongoing).

The modest apparent increase in the $K_m$ value for PAH transport at longer incubation times in kinetic experiments led us to speculate that the potency of OAT1 inhibition may be sensitive to incubation time as well. The potency with which five drugs inhibit OAT1 was tested using either 15 s or 10 min incubation time points. Drugs with a variety of inhibition mechanisms were chosen, including competitive (probenecid, indomethacin, and furosemide), mixed-type (omeprazole), and non-competitive (telmisartan) [21]. Interestingly, $IC_{50}$ values were sensitive to incubation time for some inhibitors, but not all. Compared to 15 s, when using the 10 min incubation time, there was no difference in $IC_{50}$ values for omeprazole and furosemide, indomethacin's $IC_{50}$ was higher, and probenecid's and telmisartan's were lower. The most dramatic effect was with telmisartan, which showed a 7-fold higher potency following a 10 min vs. a 15 s co-incubation with PAH. The present study is not the first to show that incubation time can influence inhibition potency toward a drug transporter. Studies with OATP1B1 and OATP1B3 reported a 5- to 20-fold increase in cyclosporin A inhibition potency following a ~30 min pre-incubation compared to a brief co-incubation with substrate [22,23]. Accordingly, in its guidance on drug interaction studies, the US-FDA suggests that sponsors of new chemical entities should consider performing a pre-incubation step when examining inhibition potential against OATP1B1 and OATPB3 [5]. However, no mention of including a pre-incubation step for other pertinent drug uptake transporters, such as OAT1, OAT3 or OCT2, is mentioned in the guidance document.

Data suggest that transport proteins alternate between outward- and inward-facing conformations during substrate translocation, and that structural rearrangements occur in transmembrane helices, leading to an alteration in the physical location of amino acids in the ligand-binding pocket [24–26]. It is possible that differences in $IC_{50}$ values, depending on incubation time, reflect differences in binding affinity of the ligand for the extracellular-vs. intracellular-facing conformation, and where there are differences, the time it takes for the drug to reach the binding pocket in its intracellular-facing conformation. While this is a possible mechanism to explain the observed results, further experimental investigation is required.

To further explore the potential mechanism for the effect of time on inhibition potency observed for telmisartan, we conducted classical inhibition studies that are done to determine if a drug is a mechanism-based (covalent-modifier) inhibitor of cytochrome P450s (CYPs). That is, we pre-incubated the cells with various concentrations of telmisartan for various time periods before examining the PAH uptake. A significant effect of time on OAT1-mediated transport inhibition was observed at each concentration of telmisartan tested. However, it was not the typical linear effect that is seen with mechanism-based inhibition of CYPs. There was a time effect from 1 to 5 min, but not thereafter. Washout experiments showed that the inhibitory effect of telmisartan was reversible, albeit slowly. Thus, telmisartan likely has a relatively slow off-rate from its OAT1-binding region, resulting in a relatively low $K_d$ value ($K_d = k_{off}/k_{on}$). Indeed, drugs with slow off-rates tend to be more potent inhibitors [27], and telmisartan is a more potent OAT1 inhibitor compared to other inhibitors [11].

## 5. Conclusions

The present study used in vitro assays with OAT1 expressed in CHO cells to show that the time of incubation with substrate influences the kinetics of PAH transport ($J_{max}$ and $K_m$ values), and the time of incubation with inhibitor can affect the OAT1 inhibition potency ($IC_{50}$ values). PBPK modeling using kinetic parameters obtained at different incubation time points showed that the time point chosen for in vitro studies impacts renal clearance predictions for PAH. Importantly, the contribution of active transport to renal tubular

secretion could be underestimated if performing kinetic experiments beyond an initial-rate time point.

**Supplementary Materials:** The following supporting information can be downloaded at: https://www.mdpi.com/article/10.3390/jox13020016/s1, Figure S1: Pharmacokinetic predictions for PAH using its clinically-observed plasma intravenous clearance value [28,29]. Figure S2: Optimization of MechKiM pharmacokinetic predictions for PAH by parameter estimation of the OAT1 RAF value [30–36]. Figure S3: Time course of recovery of OAT1-mediated [³H]*PAH* uptake by CHO-OAT1 cells after pre-treatment with telmisartan. Table S1. (A) Effect of time on OAT1 transport kinetics. (B) Effect of time on OAT1 inhibition.

**Author Contributions:** Conceptualization: R.M.P.; participated in research design: R.M.P.; conducted experiments: A.O.B.; performed data analysis: A.O.B.; wrote or contributed to the writing of the manuscript: R.M.P. and A.O.B. All authors have read and agreed to the published version of the manuscript.

**Funding:** A portion of this research was supported by the Canadian Institutes of Health Research—Nova Scotia Regional Partnership Operating Grant #129209 (CIHR) and MED-MAT-2013-8813 (NSHRF).

**Data Availability Statement:** The data presented in this study are available on request from the corresponding author. The data are not publicly available as the corresponding author is an employee of Novartis.

**Conflicts of Interest:** R.M.P. is an employee of the Novartis Institutes for Biomedical Research. A.O.B. is an employee of LabCorp. A portion of this work was completed in fulfillment of Aaron Buaben's MSc research thesis conducted in the Department of Pharmacology at Dalhousie University. The online version of this thesis can be found at: Evaluating the Optimal In Vitro Transport Conditions for PBPK Modeling of OAT1 Involvement in Renal Drug Clearance (dal.ca).

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
