# Peer review of "Incubation Time Influences Organic Anion Transporter 1 Kinetics and Renal Clearance Predictions"

_jox, doi:10.3390/jox13020016_

Round 1
Reviewer 1 Report
Buaben and Pelis describe in this manuscript the effects the incubation time has on the determination of kinetic parameters for OAT1-mediated substrate transport. Overall this is an important study that demonstrates that kinetic parameters should be determined under initial linear rate conditions. There are a few things that should be clarified.
1) In line 129, was the PAH uptake at time zero determined experimentally?
2) In the pretreatment with telmisartan did the uptake solution also contain telmisartan or not?
3) It looks as if uptake in CHO parent cells or vector-transfected control cells was not measured and subtracted from uptake into CHO-OAT1 cells. Why not? In Figure 1. it is quite obvious that the linear portion does not go through zero, probably because of unspecific binding that could be determined and corrected for. Does this not lead to an overestimation of uptake, especially at higher concentrations?
4) In the DDI experiment, the IC50 value was similar to what the authors determined at 10 minutes. What would the simulation results be with the IC50 measured at 15 seconds?
5) In the discussion, active transport and passive diffusion are mentioned as contributors to total uptake. Should it not be protein-mediated or transporter-mediated transport instead of active transport? Is OAT1 indeed an active transporter? To keep this discussion more general, it might be better to not use active transport but protein-mediated transport.
6) On a similar note, passive diffusion (I mean simple diffusion that does not rely on OAT1) should be measurable with the CHO parental cells. It could be subtracted from the total uptake to yield net transporter-mediated uptake. This would probably make things a lot easier.
7) I am not sure I fully agree with the statement in line 356 that at 45 minutes, PAH uptake is attributed to passive diffusion. Do you get the same uptake at 45 minutes with the parental CHO cells?
Minor:
In line 512, the reference WD, S. (1986) should be corrected to Stein, W.D. (1986)
Author Response
Thank you for your careful review of our manuscript. We have adjusted the text according to your feedback.
1. No, the earliest time point that uptake was conducted was at was fifteen seconds. At this time point total 3H radioactivity taken up into CHO parental vs. CHO-OAT1 cells is typically 100 vs. 1000 dpm.
2. No, the uptake solution did not include telmisartan. Telmisartan was only included in the pre-incubation step. We have included the following statement on Pg. 5. “In these experiments the cells were pre-incubated with telmisartan, but telmisartan was not included in the uptake solution.”
3. No, we did not measure uptake in the control cells as performing radiolabel displacement type studies (i.e., uptake of a tracer level of 3H with increasing concentrations of unlabeled compound) in control cells gives a similar background level of 3H uptake regardless of the unlabeled compound concentration. The displacement kinetic analysis (line 142) incorporates the parameter C, which accounts for: “C is defined as the non-saturable component of [3H]PAH uptake that is most likely due to factors such as non-specific binding, incomplete rinsing, and passive diffusion.” (as stated on line 144-146). That is, we have accounted for the non-saturable component of uptake in our determination of the kinetic parameters Jmax and Km.
4. There was an approximate 2-fold difference in probenecid IC50 at 10 min compared to 15 sec (5.1 vs. 9.3 uM). When we simulate the DDI using the IC50 determined at15 sec the DDI magnitude was similar to using the IC50 at 10 min (1.8 vs. 1.5-fold difference in AUC).
5. We agree with this recommendation. Instead of using the term ‘active’ transport we have changed to ‘transporter-mediated’.
6. This has been addressed in point 3 above.
7. We agree completely with your comment. Please see revised text.
- Minor, Stein W.D. fixed.
Reviewer 2 Report
This manuscript describes the influence of incubation time of transporter studies with involving OAT1 on kinetic parameters, showing higher Km and Ki values at steady-state as compared to initial rates. They then used the commercially available Simcyp Simulator version 15 (Certara, Inc) to do physiologically based pharmacokinetic (PBPK) modeling. The novelty level is moderate, and the results can be used to benchmark parameters of for OAT1 for PK modeling.
Author Response
Thank you for reading our manuscript
Reviewer 3 Report
Thee manuscript is very well written and detailed. Sound scientific approached have been used. The study design for wet lab experiments has been well thought and required assumptions have been considered for modeling purposes. The manuscript is eligible for publication upon incorporation of a few comments noted below.
1) Have you compared the predicted profile from both initial rate and in vitro SS to a multidosing regimen of paramino hippuric acid?
2) In the discussion, for better awareness to the readers can you comment on how would the intial rate Clint values approach prediction compare with multi-dosing profile when the drug may have reached steady state (as opposed to single dose).
Author Response
Thank you for your review of our manuscript.
- We have not compared the in vitro kinetic parameters conducted at initial rate vs. steady state using a multidosing regimen as the available clinical data for comparing predicted vs. observed was primarily single dose intravenous.
- Regardless if the dosing is single vs. multiple we expect that transporter-mediated renal clearance would be under-estimated if using kinetics at steady state as opposed to initial rate. We have noted this in the discussion section.
Round 2
Reviewer 1 Report
The authors have addressed my comments/suggestions.